# Load Balancing Scheme for Effectively Supporting Distributed In-Memory Based Computing

**Kyoungsoo Bok, Kitae Choi, Dojin Choi, Jongtae Lim and Jaesoo Yoo \*** 

School of Information and Communication Engineering, Chungbuk National University, Chungdae-ro 1, Seowon-Gu, Cheongju, Chungbuk 28644, Korea; ksbok@chungbuk.ac.kr (K.B.); hissker@hanmail.net (K.C.); mycdj91@chungbuk.ac.kr (D.C.); jtlim@chungbuk.ac.kr (J.L.)

\* Correspondence: yjs@chungbuk.ac.kr; Tel.: +82-43-261-3230

**Abstract:** As digital data have increased exponentially due to an increasing number of information channels that create and distribute the data, distributed in-memory systems were introduced to process big data in real-time. However, when the load is concentrated on a specific node in a distributed in-memory environment, the data access performance is degraded, resulting in an overall degradation in the processing performance. In this paper, we propose a new load balancing scheme that performs data migration or replication according to the loading status in heterogeneous distributed in-memory environments. The proposed scheme replicates hot data when the hot data occurs on the node where a load occurs. If the load of the node increases in the absence of hot data, the data is migrated through a hash space adjustment. In addition, when nodes are added or removed, data distribution is performed by adjusting the hash space with the adjacent nodes. The clients store the metadata of the hot data and reduce the access of the load balancer through periodic synchronization. It is confirmed through various performance evaluations that the proposed load balancing scheme improves the overall load balancing performance.

**Keywords:** distributed in-memory; load balancing; data migration; replication; hot data

## 1. Introduction

As the use of digital devices, such as tablet PCs and smartphones, has increased in recent times with the rapid growth of social media applications, such as Twitter and Facebook, various forms of digital data have also increased exponentially in our everyday life. As a result, big data technology has emerged to effectively manage and process large volumes of data that cannot be handled in a traditional manner [1–3]. Big data can introduce new meaning by processing and analyzing large amounts of data, which could not be found in the existing data. This is being used in various fields, such as trend analysis, marketing, and decision making [4,5].

Distributed storage processing techniques, such as Hadoop, have been used to process large amounts of data beyond the processing limits of the existing storage and processing systems [6–8]. Hadoop is a representative open-source software framework for the distributed storage and processing of large amounts of data. As Hadoop stores and processes large amounts of data in the disks of distributed nodes, continuous disk input and output occur, making real-time processing impossible [9–11]. Furthermore, when input and output are concentrated on a specific node, a bottleneck occurs, and the overall processing speed is lowered. To address such disk input and output problems, distributed in-memory technology has emerged, allowing the distributing, storing, and processing of data in the memory to have a fast access speed [12–14]. Distributed in-memory technology is being widely used in applications that process large amounts of data in real-time [15–18]. An example of a representative in-memory processing technology is memcached [19–21].

Memcached is a key-value-based memory cache that is frequently used in applications that provide online real-time services, including Facebook, Twitter, Reddit, and YouTube. Memcached reduces the storage access to the back-end databases by directly storing user data requests in the distributed in-memory [22,23]. In addition, as it uses the memory of the distributed nodes as a single storage, a large amount of data can be stored in the memory and can be used subsequently [24,25]. Memcached has attracted much attention due to its high usability, and studies are being actively conducted to further improve its performance. As Memcached operates in distributed environments, load imbalance may occur among nodes. In other words, in the distributed in-memory environments, when requests are concentrated on a specific node or the use of specific data is focused, a problem of load increase on a specific node occurs. Such a load imbalance among the nodes degrades the overall system response time and the network performance [26–31].

To address such a load imbalance among nodes in the distributed in-memory environments, studies have been conducted using ring-based hashing schemes [32–37]. The general ring-based hashing schemes adjust the loading by replicating the data to other nodes, or by migrating the data through a hash space adjustment. The study in [33] calculated the load on a node using the hit rate and the usage rate, and performed load balancing by adjusting the hash space. If hot data exists in a specific node, however, the hit ratio and the usage rate will increase, and many hash spaces must be adjusted. In other words, the occurrence of hot data significantly increases the data migration cost. The studies in [34,36] proposed a scheme of distributing the load concentrated on one node by replicating the hot data that causes a large load to another node. When load balancing is performed considering only the hot data, however, it is not possible to solve the situation where a load occurs on a node without the hot data. Moreover, in heterogeneous environments, when load balancing is performed without consideration of the memory size, frequent data migration occurs in a node with a relatively small memory size. As such, it is difficult to actually apply the existing load balancing schemes because they perform load balancing only for a specific situation, and do not consider both the load of the node itself and the data load.

We propose a load balancing scheme in distributed in-memory [37]. The authors of [37] proposed the concept of load balancing using data replication and data migration in a distributed in-memory environment. However, Ref. [37] did not provide initial data distribution techniques in heterogeneous distributed in-memory environments and did not check the load status of an overloaded node after data replication and data migration. In addition, load balancing due to the addition and removal of nodes was not performed for distributed in-memory processing. In this paper, we propose a new load balancing technique that extends the existing scheme [37] in heterogeneous distributed in-memory environments. The proposed scheme distinguishes the load status according to whether hot data is generated and performs data replication or data migration using a load balancer. It also performs load balancing by adjusting the hash space according to the addition and removal of the nodes. The clients maintain up-to-date metadata by periodically synchronizing the metadata information about accessible data with the load balancer. The main contributions of this paper over [37] are as follows:

1. The proposed scheme takes into account the performance of the nodes in a heterogeneous distributed in-memory environment, distributes the initial data, and determines the load status of the nodes through the load balancer.
2. This paper presents detailed algorithms for data replication and migration according to the load status of the node and proposes a technique to determine the load status of the node according to the load distribution.
3. The proposed scheme adjusts the hash space by considering the load status of the node for load balancing when adding or removing the node.
4. The proposed scheme reduces access to the load balancer because the client maintains metadata about the hot data through periodic synchronization with the load balancer.

The rest of this paper is presented as follows. Section 2 describes the related works, and Section 3 describes the proposed scheme for load balancing among the nodes in the distributed in-memory environments. Section 4 verifies the excellence of the proposed scheme through performance evaluation and analysis, and Section 5 describes the conclusion of this paper.

## 2. Related Work

Hwang et al. proposed an adaptive hashing scheme to support load balancing in distributed in-memory caching systems [33]. Since the stored data size and data access frequency are different, each node has a different load. When a load imbalance occurs, the cost of the node is calculated by using a hit rate and a usage rate of each node and then the hash space is adjusted to balance the loads across multiple cache servers. When a new node is added, the new node takes over exactly half of the hash space from the overloaded node in the counter clockwise direction to assign a balanced hash space to the new node. In addition, when the node is removed, the two immediately adjacent virtual nodes will divide the hash space of the removed node.

Zhang et al. proposed a load balancing scheme that extracts hot data that causes a large load and replicates it to other nodes [36]. To distribute the load, the hot data is detected using the lossy counter and the hot data is replicated to the more efficient node. When a client requests data, the load balancer accepts the client's request and sends it to the hot spot detector. The hot spot detector continuously analyzes the client's request using the lossy counter, which separates keys into groups with similar request rates. The load balancer also maintains a forward table for the separate hot data in the key director. If the user's requested data is hot data, the data can be accessed quickly through the forward table.

Lu et al. developed a distributed caching middleware called R-Memcached, which incorporates a replication technique into Memcached to prevent a large number of concurrent requests from sending to a single node [34]. When the data is stored, the replica is stored in the clockwise nodes using a ring-based hash. Hot data is replicated to other nodes to solve the load imbalance problem that may occur at a particular node. The round-robin technique is used to prevent the data request from concentrating on a particular node. After R-Memcached adds a replica technique, the probability of data loss is reduced.

The existing schemes that have been proposed for load balancing cannot address the problem of a load imbalance among nodes that occurs in various situations because they distribute the load while considering only specific situations. Distributed management schemes using data replication extract the hot data that causes a severe load on a node, and distribute the hot data to other nodes, thereby performing load balancing. When the hot data is replicated to the neighbor nodes, however, the hot data can be stored again in one of those nodes when a node with the replicated hot data is removed, resulting in load concentration again. In addition, memory space is wasted due to the overlapping of memory storage. As no load balancing scheme has been proposed for the case where hot data does not exist, proper load balancing cannot be performed if the hot data does not exist. The load balancing management scheme that uses data migration calculates the cost based on the hit rate and the usage rate for each node, adjusts the hash space, and migrates the data, thereby performing load balancing. When the node is overloaded due to the hot data, however, the cost of the node is considerably increased, many hash spaces need to be adjusted, and a high data migration cost is required. Furthermore, as the load is distributed by adjusting the hash spaces of the neighboring nodes, appropriate load balancing cannot be performed if the neighboring nodes are under heavy loads.

To address the problems of the existing schemes, a novel scheme is required that can address the load imbalances caused by various situations. First, the overloaded node needs to be found using the status information of the nodes, and the existence of hot data in the overloaded node must be confirmed to identify the cause of the node overload. As the existing replication schemes replicate hot data to the neighbor nodes of the overloaded node, the hot data can be stored again in one of those nodes when a node with the replicated hot data is removed. Therefore, a scheme is required

that can address the problem of overlapping data storage when a node is removed by replicating the hot data evenly to the other nodes. When no hot data exists in the overloaded node, and the data is migrated by adjusting the hash spaces of the neighbor nodes, a large adjustment of the hash spaces leads to a high migration cost, and the overload problem cannot be addressed when the neighbor nodes are under heavy loads. Therefore, when there is no hot data, a load balancing scheme is required, which minimizes the data migration considering the loads of the nodes. In an environment in which the memory sizes are different, a node with a small memory size can be easily filled with data, and thus, data change may occur frequently. Therefore, a scheme is required which can store the data considering the memory size.

## 3. Proposed Load Balancing Scheme

### 3.1. Architecture

In the distributed in-memory environments, when requests are concentrated on a specific node, an overload problem occurs, and the overall system performance is degraded. Therefore, a load balancing scheme is required to address the load imbalance by distributing the load of the overloaded node. In the existing load balancing schemes, the load is distributed using a scheme that replicates or migrates the data considering only a specific situation in which the node overload occurs. When the data is replicated to the neighbor nodes, the data can be stored again in one of those nodes when a node with the replicated data is removed, resulting in the overload problem again. In addition, if hot data exists in a node when the data is migrated, the hash spaces are significantly adjusted, and a large amount of data is migrated. In an environment where the memory sizes are different and when a node with a small memory manages a large hash space, it must manage a large amount of data. However, the storage is limited due to the memory size, resulting in frequent data replacements.

In this paper, a load balancing scheme considering the load condition of the nodes is proposed to distribute the load in the distributed in-memory environments. The proposed scheme extracts the overloaded nodes and hot data based on the node and the data loads, and distributes the load through a data replication and migration process. In an environment where the memory sizes of the nodes are different, load balancing is performed by adjusting the initial hash spaces according to the memory size. If the hot data exists when a node is overloaded, the hash spaces are evenly divided, and the hot data is replicated to a node with a low load for load balancing. If there is no hot data, the data of the node with the lowest load is distributed considering the loads of the neighbor nodes, and load balancing is performed by selecting the node as the predecessor node of the node with the highest load and by adjusting the hash space. In distributed environments, nodes can be added or removed. In a situation where a node is added, the new node is added as the predecessor node of the overloaded node in order to distribute the load of the overloaded node. In a situation where a node is removed, the hash spaces are adjusted and the data is migrated considering the loads of the successor node, and the predecessor node of the node to be removed.

Figure 1 shows the proposed load balancing system. A load balancer operates in a node acting as a central server, distributes the data to the nodes for storage, and performs load balancing for the distributed memory. When the user's data request is received, the load balancer determines whether the requested data exists in the distributed memory, and delivers the data node information. If there is no user requested data in the memory, the load balancer loads the data from the disk into the memory, and transfers the data to the user. If the data requested by the user is not stored in the memory of the node, the data distributor designates a node for the storage of the data and stores it. The hot data manager collects and separately manages the hot data information generated from the nodes, and synchronizes the metadata of the hot data through periodic communications with the clients.

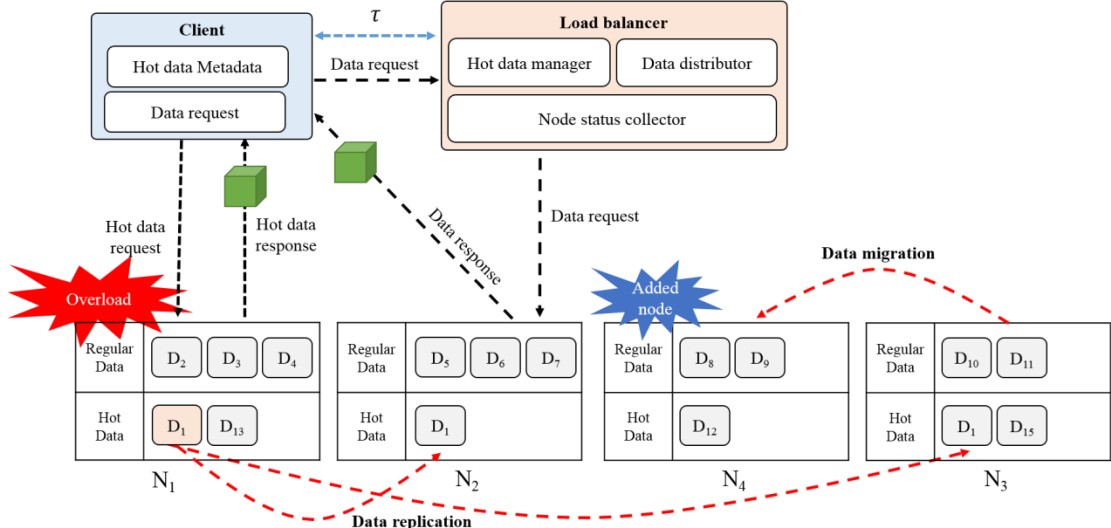

**Figure 1.** Overall architecture.

When an overload occurs at a node, the node sends its current status information to the load balancer. The load balancer that receives the information of the overloaded node collects the load status of all the current nodes based on the access frequency, and stores the status information to distribute the load. The load balancer also collects and stores hot data information from all the nodes. The load balancer distributes the load through a hot data replication and data migration process, based on the load information and the hot data information received from each node. If there is hot data in the overloaded node, the load balancer replicates the hot data to another node, and manages the metadata information of the replicated hot data. If there is no hot data, the load balancer adjusts the hash spaces based on the load on the node, maintains the adjusted hash space of the node, and accesses the node using the adjusted hash space when a data request is received.

### 3.2. Initial Data Distribution

In the distributed in-memory environments, when nodes are added or removed, the data must be redistributed to the distributed nodes. In general, in distributed environments, however, all data of the existing nodes must be redistributed when the nodes are added or removed. The ring-based hashing scheme, however, does not redistribute all the data when the nodes are added or removed. It only redistributes some data by adjusting the hash space to be managed by other adjacent nodes. This reduces the overall load on the system, thereby increasing the data redistribution efficiency in the distributed environments. Therefore, the proposed scheme is based on the ring-type chord [38,39]. In the distributed in-memory environments, the memory size of each node may be different. In this instance, when a node with a small memory size manages a large hash space, a large amount of data must be stored in the node, requiring more frequent data replacement due to the limited memory. In this case, the data replacement causes considerable cost, and the performance of the node may be degraded. Therefore, the proposed scheme distributes the initial nodes considering the memory of the nodes, and addresses the problem caused by data replacement.

Figure 2 shows an example of the identifier space of the ring-type chord. $N_1$ to $N_5$ represent the nodes constituting the identifier space. The adjacent node located in the counterclockwise direction of the node, $N_i$, is referred to as the predecessor of the node, $N_i$, and is expressed as a predecessor ($N_i$). The adjacent node located in the clockwise direction of the node, $N_i$, is referred to as the successor of the node, $N_i$, and is expressed as a successor ($N_i$). The chord scheme hashes the nodes and data, and maps them to a single value on the ring. The node manages the hash space between itself and its predecessor node, and stores the data of the corresponding hash value in the node. If a node is

approached by comparing the hash values to read the value stored in the node, the requested value is returned.

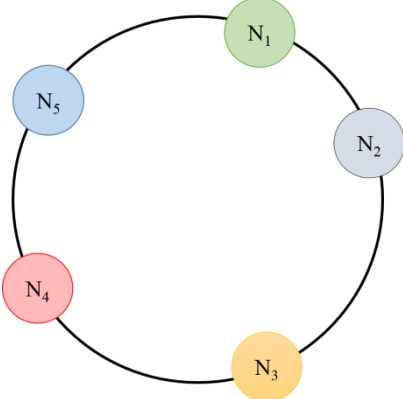

**Figure 2.** Identifier space of the chord.

The proposed scheme uses a modified chord applying a ring-based hashing scheme in order to distribute the initial nodes in environments with different memory sizes. In environments with different node sizes, the initial nodes are distributed considering the memory size of each node. In the existing distribution scheme, the nodes are distributed by hashing their unique values. In this case, the nodes with a large memory may manage a small hash space while the nodes with a small memory may manage a large hash space. Therefore, through the distribution of the initial nodes according to their memory sizes, the memory sizes are fully used, and the problem of frequent data replacement is addressed. Figure 3 shows the initial node distribution that consists of five nodes and uses a basic hash scheme. The number displayed on each node represents the memory size. When the nodes are distributed, the basic ring-based hashing scheme distributes the nodes by hashing information that can identify the node, such as in IP. In this instance, $N_5$, with a smaller memory size than $N_4$, may manage more hash spaces than $N_4$. Therefore, the memory of the node, $N_5$, is easily filled and cache replacement frequently occurs.

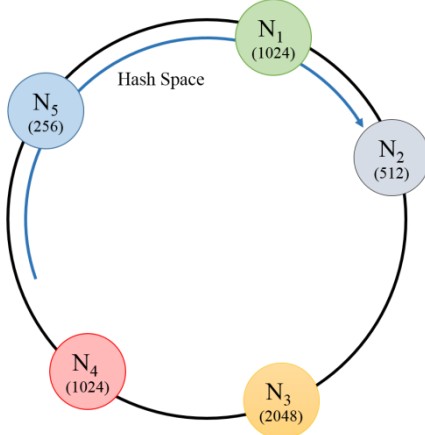

**Figure 3.** Initial nodes using the existing chord scheme.

Figure 4 shows the initial node distribution of the proposed scheme with five nodes. The proposed scheme adjusts the hash spaces considering the memory sizes based on the existing initial node distribution. For example, as $N_1$ has a larger memory size than $N_2$, the hash space of $N_1$ is adjusted to manage more data. In addition, as $N_2$ has a smaller memory size than $N_3$, the hash space of $N_2$ is not adjusted.

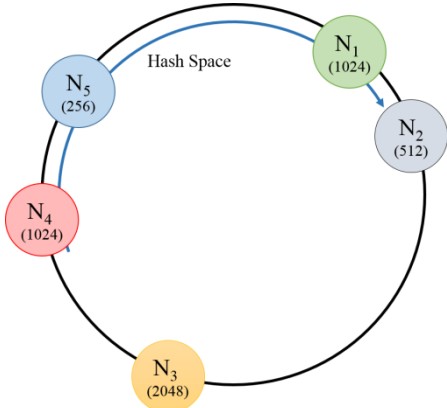

**Figure 4.** Initial node allocation using modified chord scheme.

Equation (1) is used to adjust the hash space of each node considering the memory size. If $N_i$'s memory size is greater than that of $N_i$'s successor node, $N_{i+1}$, the size of the hash space using Equation (1) is increased. Here, $H_i$ is the hash space of $N_i$ before adjustment and $H_{i+1}$ is the hash space of the node, $N_{i+1}$, before adjustment, and $MH_i$ is the adjusted hash space of the node, $N_i$:

$$MH_i = \frac{H_i + H_{i+1}}{2} \tag{1}$$

Figure 5 shows the proposed initial node allocation algorithm. We assume $H_i$ and $H_{i+1}$ are the hash space of the node, $N_i$, and the successor node of $N_i$, respectively. Node $N_i$ compares the memory size of its own memory with that of the successor node. If the memory size of node, $N_i$, is larger than the memory size of the successor node, the proposed scheme increases the hash space using Equation (1) to store more data. If the memory size of the successor node is larger than its own memory size, it does not adjust the hash space.

---

***Initial_hash space_allocation***

01 :　$H_i$= *hash space of $N_i$;*

02 :　$H_{i+1}$= *hash space of successor node of $N_i$;*

03 :　*for(each node $N_i$)*

04 :　　*if (memory size of successor($N_i$ )<memory size of $N_i$)*

05 :　　　*return $MH_i$=($H_i$+$H_{i+1}$)/2;*

06:　　*else*

07:　　　*return previous hash space;*

---

**Figure 5.** Initial node allocation algorithm.

### 3.3. Load Balancing Processing

In distributed environments, the data is distributed to multiple nodes and is shared. As the data in each node is different in distributed environments, the data requests may be concentrated on a specific node, and the performance may be degraded due to an increase in the load. In many cases, a node overload is caused by the occurrence of hot data where loading is concentrated on a specific data, or the limited node performance. Therefore, in distributed environments, it is important to identify the cause of a node overload, distribute the load, and prevent the node from overloading. To address the node overload problem that can occur in various situations, a load balancing scheme is required according to the situation.

The proposed method largely distributes the load through data replication and migration. It determines whether the current node is overloaded using the load status of the node, and identifies the hot data using the node and data loads. If hot data exists in the overloaded node, the hot data is replicated to another node to distribute the load. If there is no hot data, the hash space of the node with a low load is adjusted to reduce the load of the overloaded node and migrates the data of the overloaded node. Figure 6 shows the data replication and migration procedure. The nodes are distributed using the ring-based hashing scheme, and each node holds data according to its hash space. When an overload occurs in a node, the load is distributed through the replication or migration of the data. The data replication process distributes the load by replicating the hot data that causes a large load to another node. In this instance, the hash spaces are divided as evenly as possible based on the hash ring, and the hot data is replicated to the nodes without overload. The overlapping hot data can be stored in one node when a node with the replicated hot data is removed, preventing any problems. In addition, the hash spaces of the overloaded node and the underloaded node are adjusted to migrate the data and distribute the load. For example, if $N_1$ is overloaded and has $D_1$ (hot data), it replicates the hot data to the nodes, $N_3$ and $N_5$, that are not overloaded. As the overloaded $N_4$ does not have hot data, the hash spaces are adjusted and $D_{14}$ is migrated from $N_4$ to $N_5$.

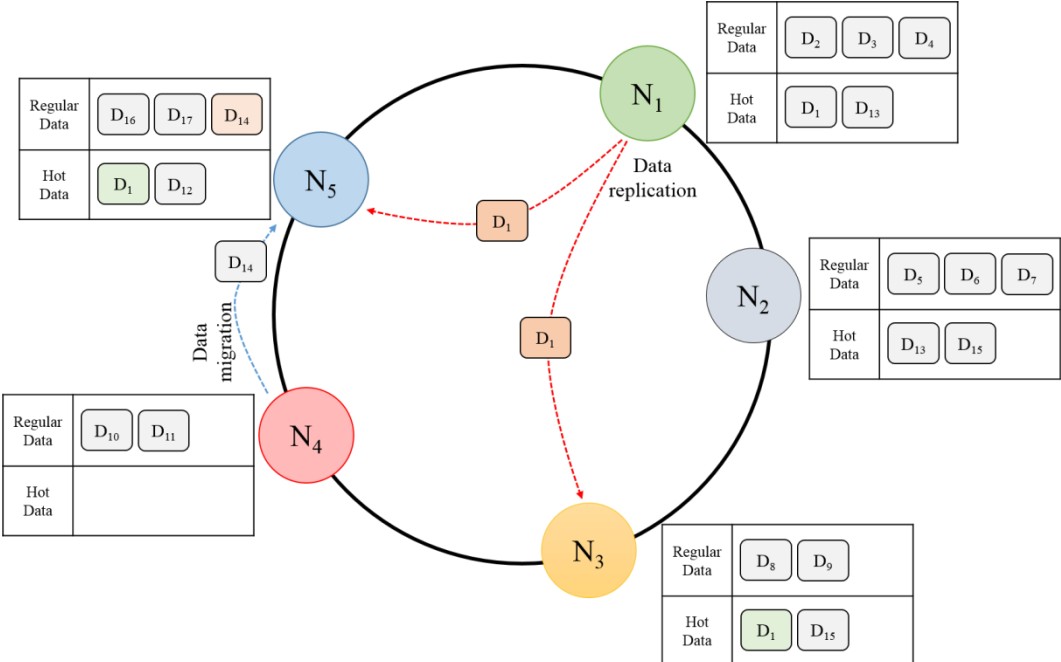

**Figure 6.** Date replication and migration procedure.

The performance of a node is reduced when the amount of load that can be processed by the node is exceeded. Therefore, in such a case, the node is identified as an overloaded node and the load is distributed. If the amount of the load that can be processed by a node exceeds the threshold value as shown in Equation (2), the node is identified as an overloaded node. Here, $N_{OL}$ represents the threshold value of the overloaded node, $\alpha$ is a parameter that determines the amount of the load, and $N_S$ is the load status of the current node:

$$N_{OL} \geq \alpha \times N_S \tag{2}$$

If an overloaded node occurs, the hot data must be determined for data replication. The hot data is determined based on the loads of data and node as shown in Equation (3). Here, $NL_i$ is the whole load of node $N_i$, $DL_k$ is the load of data $k$ stored in node $N_i$, and $\beta$ is the threshold value. The hot data is determined based on the ratio of the load generated by the data to the total load on the node. When the

load generated by the data exceeds the threshold value of the total load, such data is regarded as hot data:

$$\frac{DL_k}{NL_i} \geq \beta \qquad (3)$$

Figure 7 shows the load balancing processing procedure algorithm. If a particular node is an overloaded node for a certain period of time, then data replication or data migration is performed depending on whether hot data exists on the overload node. If hot data exists on an overload node, then replication to the hot data is performed and otherwise the hash space is adjusted to migrate the data. If a node performing the service in a distributed environment is added or removed, the hash space to migrate the data is adjusted. If a new node is added, the proposed scheme sets the added node to the predecessor node of the overload node and adjusts the hash space of the overload node to migrate data to the new node. If an existing node is removed, the hash space is adjusted according to the load status of the predecessor node and the successor node, and the data stored on the removed node is migrated to the predecessor node and the successor node.

---

*load_balancing processing*

---

*01 :  if (the node is overloaded for a certain time interval)*

*02 :   if (there is hot data in the overloaded node)*

*03 :    replicate hot data to the node with a low load ;*

*04 :   else*

*05 :    migrate the data through hash space adjustment;*

*06 :  if (a node is added or removed)*

*07 :   if (a node is added)*

*08 :    set a new node as the predecessor node of the overloaded node;*

*09 :    migrate the data to the new node through hash space adjustment;*

*10 :   else*

*11:     distribute the hash space of the removed node according to the load status of the predecessor node and the successor node;*

*12:     migrate the data of the removed node to the predecessor node and the successor node;*

---

**Figure 7.** Load balancing processing procedure algorithm.

### 3.4. Load Balancing through Data Replication

Hot data refers to the data that is frequently used and causes a large load on a node. As the hot data greatly increases the load on a node, the load balancing processing scheme that uses such data is important. The load imbalance due to the hot data can be solved by replicating such hot data to other nodes to distribute the load of the hot data to other nodes. When the hot data is replicated to the neighbor nodes, however, if one of those nodes is removed, the hot data can be stored again in another node, causing an overload again. Therefore, the proposed scheme addresses the problem of such overlapping storage of hot data. It divides the hash spaces evenly based on the hot data and stores one replica in each divided hash space considering the load on the corresponding node. This can address the problem of overlapping data storage when a node is removed.

Figure 8 shows the procedure of hot data replication. The entire hash space is evenly divided according to the number of the hot data to be replicated. In each divided hash space, the nodes are checked for overload sequentially from the one with the low hash value, and the hot data is replicated to the node without overload. In this instance, the basic number of replicas is three, including the original data. Therefore, the entire hash space is divided into three ranges based on the hash value of the original data. The original data is retained in the previously stored node. The first replica is stored in $R_2$ after examining the loads in a sequence. The examination in $R_2$ starts from $N_6$, the first node of $R_2$. If $N_6$ is overloaded, the next node, $N_7$, is examined. If $N_7$ is not overloaded, the second hot data

is stored in $N_7$. For the second replica, $N_{10}$, the first node of $R_3$, is examined. If it is not overloaded, the replica is stored in $N_{10}$.

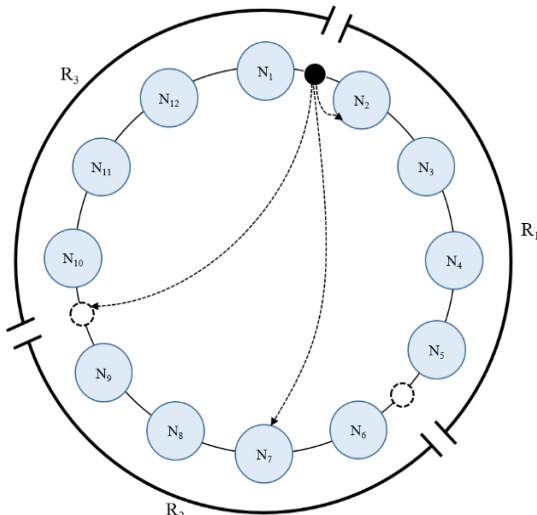

**Figure 8.** Hot data replication.

When a node is removed, the data of the node is migrated to the neighbor node. If the hot data is replicated to the consecutive neighbor nodes, and a node with the replicated hot data is removed, the hot data can be stored again in another node. In this case, as two hot data replicas are stored in one node, the memory space is wasted and the load is also concentrated. Therefore, if the hot data is stored in the hash space and divided as evenly as possible considering the loads of the nodes, the overlapping of data storage in one node can be prevented even if a node with a replica is removed.

Equations (4) and (5) are used to obtain the hash space for the data replication. Equation (4) calculates the hash value that is used to obtain the hash space for data replication. Here, $HRR_i$ is the hash value that is used to obtain the replication range, $H_{Max}$ is the entire hash space, $R_N$ is the number of replicas, and $H_{HD}$ is the hash value of hot data. Equation (5) calculates the hash space, $HR_i$, for data replication using the hash value calculated through Equation (4). Here, $HR_i$ is the hash space, and $HRR_i \sim HRR_{i+1}$ are the hash ranges obtained using the hash value calculated through Equation (4). For hot data replication, the ranges in the hash space are determined using the calculated value, and one data is replicated to each range to prevent the data from being replicated to the consecutive neighbor nodes:

$$HRR_i = H_{Max} \times \frac{i}{R_N} \times H_{HD} \tag{4}$$

$$HR_i = HRR_i \sim HRR_{i+1} \tag{5}$$

After the load balancing through data replication, it is necessary to examine the load status of the overloaded node, and confirm whether the distribution has been performed well. When an overload occurs and the load is distributed, the load of the overloaded node decreases as shown in Equation (6). Here, $NL_i$ is the load of node $N_i$ before replication, $n$ is the number of replicas, $m$ is the number of hot data in node $N_i$, $HDL_k$ is the load of hot data $k$ in node $N_i$, and $RNL_i$ is the load of node $N_i$ after the replication of hot data. After the data replication, the load status of the node is examined through the calculated load value, and then the load status of the node is determined based on the calculated value:

$$RL_i = NL_i - \frac{n-1}{n} \sum_{k=1}^{m} HDL_k \tag{6}$$

### 3.5. Load Balancing through Data Migration

An overloaded node may occur due to uniform access to the data held by the node rather than frequent access to its specific data. In this case, the load can be distributed by adjusting the hash spaces and by performing data migration. In the proposed scheme, the node with the lowest load is selected and its hash value is deleted. In this instance, the hash space managed by the underloaded node is adjusted and distributed by considering the loads of the predecessor node and the successor node. The underloaded node with the deleted hash value is assigned a hash value as the predecessor node of the node with the largest load node to reduce the load of the overloaded node.

Figure 9 shows the procedure of deleting the hash value of the underloaded node with the lowest load. To remove the underloaded node from the distributed environment, the managed hash space is distributed considering the loads of the predecessor node and the successor node. The hash space is distributed to the predecessor node and the successor node as the load increases significantly for the successor node if it manages all the hash space. For example, if $N_3$ is a node with the lowest load, the hash value of $N_3$ is deleted and the hash space and data of $N_3$ are distributed to the predecessor node, $N_1$, and the successor node, $N_2$.

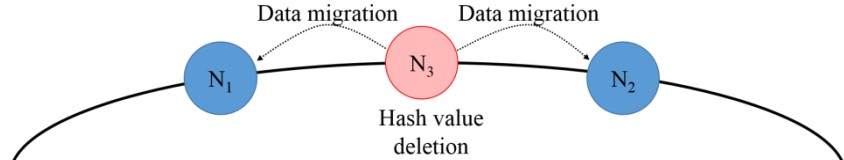

**Figure 9.** Deleting the hash value of an underloaded node.

Equation (7) is used to delete the hash value of an underloaded node with the low load in a distributed environment. The loads of the neighbor nodes on both sides of the underloaded node are compared and then the hash space of the underloaded node is distributed to those nodes. Here, $H_p$ is the hash space of the predecessor node, $N_p$; $NL_i$ is the load on node $N_i$; $NL_j$ is the load on the adjacent node, $N_j$, of node $N_i$; and $|S_k|$ is the hash space of the node to be removed. For example, if the hash space managed by $N_3$ is 501–1000, the load on $N_1$ is 20, and the load on $N_2$ is 30; $N_1$ additionally manages the hash space of 501–800 among the hash space of $N_3$, and $N_2$ additionally manages the hash space of 801–1000 for the load balancing:

$$H_p = \frac{NL_j}{NL_i + NL_j} \times |S_k| \tag{7}$$

Figure 10 shows the added underloaded node with the deleted hashed value for reducing the load on the node with the highest overload. The underloaded node is assigned a hash value as the predecessor node of the overloaded node with the highest load node to reduce the load. If $N_2$ is an overloaded node, the underloaded node, $N_3$, with a deleted hash value is assigned a hash value as the predecessor node of $N_2$, and the hash space and data are distributed for load balancing.

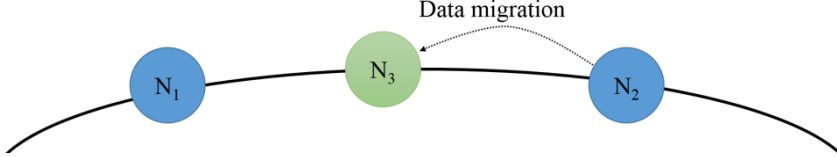

**Figure 10.** Assigning a hash value to the underloaded node.

When an underloaded node is assigned a new hash value, it manages part of the hash space of the existing node and migrates the data. Equation (8) is used to adjust the hash range when the removed underloaded node is added. Here, $H_k$ is the hash value of the added underloaded node, $N_k$, and $|N_j|$ is

the hash range of the node, $N_j$, with the highest load. For example, if $N_2$ is the node with the highest load and manages the hash range of 0–1,000, the underloaded node, $N_3$, is added as the predecessor node of $N_2$ and manages 0–500, half of the hash space of $N_2$. The load of the existing overloaded node is reduced by the newly added node, which manages part of the hash space of the node with the highest load:

$$N_k = \frac{|N_j|}{2} \tag{8}$$

After the load balancing through data migration, it is necessary to examine the load status of the overloaded node and confirm whether the distribution has been performed well. Equation (9) calculates the load of the overloaded node after the data migration. The load of the overloaded node has been reduced by the load of the migrated data. Here, $MNL_i$ is the load of the node, $N_i$, after data migration; $NL_i$ is the load of the node, $N_i$, before data migration; $k$ is the number of the migrated data; and $DL_m$ is the load of the migrated data, $m$. The load status of the node is examined through the load value calculated after the data migration, and then, the load status of the node is determined based on the calculated value:

$$MNL_i = NL_i - \sum_{m=1}^{k} DL_m \tag{9}$$

### 3.6. Load Balancing by Node Addition and Removal

In distributed environments, when a new node is added or a node is removed due to a failure, the data must be redistributed to the distributed nodes. If all the data is redistributed due to the addition or removal of a node, a load occurs on the entire system, resulting in delays in processing user requests. When a node is added or removed, the ring-based hashing scheme does not redistribute all data, but redistributes only some data by adjusting the hash space that needs to be managed by other neighbor nodes. This reduces the overall system load.

In the proposed scheme, the hash spaces are adjusted by considering the load status of the nodes when a node is added or removed using the ring-based hashing scheme. When a node is added, the new node is added at the position of the predecessor node of the overloaded node to reduce the load on the node with the highest overload. When a node is removed in the existing scheme, the successor node of the removed node manages the hash space of the removed node. In this case, the load on the successor node increases significantly. In the proposed scheme, the hash space of the removed node is distributed to the predecessor node and the successor node considering their load status.

Figure 11 shows the procedure of adding a new node. The added node is added as the predecessor node of the node with the highest load among all the nodes, thereby reducing the load of the node with the highest load. For example, if $N_3$ is newly added, it is added as the predecessor node of $N_2$ with the highest load, and it reduces the load on $N_2$ by migrating the hash space and data. The new node has 50% of the hash space of the node with the highest load, and the data contained in the hash space of the new node is migrated from the existing node. The hash value of the newly added node is calculated using Equation (7).

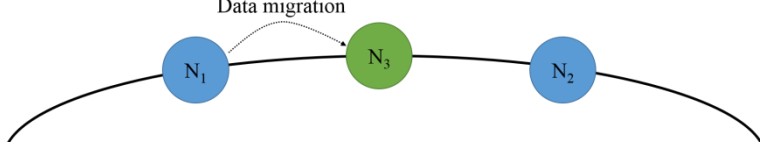

**Figure 11.** Node addition.

Figure 12 shows the procedure of removing a node. If the node is removed, the successor node manages the hash space of the removed node. The successor node of the removed node then manages more hash space, and its amount of data increases, resulting in an increased load. To address this

problem, the hash spaces of the predecessor node and the successor node of the removed node are adjusted considering their load status. When the node, $N_3$, is removed, $N_2$, the predecessor node of $N_3$, and $N_1$, the successor node, divide and manage the hash space of $N_3$. When a node is removed, the hash values of the predecessor node and the successor node are calculated using Equation (6).

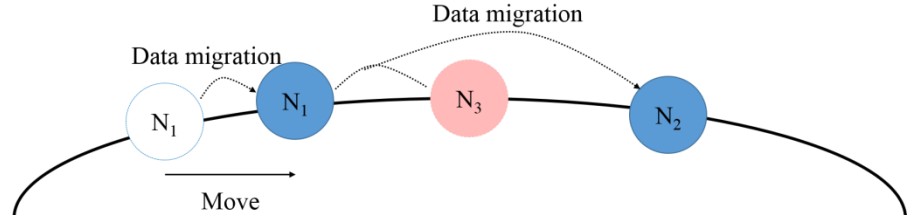

**Figure 12.** Node removal.

### 3.7. Metadata Synchronization of Hot Data

In the distributed environments, the nodes that hold the data are accessed through the load balancer. If the data requests an increase, however, the load on the load balancer increases and its performance is degraded, lowering the overall system performance. In addition, as clients request data to the nodes, the data used by the nodes are changed, and the metadata information of the hot data managed by the load balancer is also modified. In this instance, if the metadata information of the hot data held by the clients differs from that of the load balancer, it is highly probable that incorrect accesses are performed upon data requests. The probability of incorrect accesses is lowered by organically adjusting the communication cycle ($\tau$) of each client, and by updating the hot data metadata information of the clients. As each client directly accesses the nodes without going through the load balancer, the time required to access the hot data is reduced.

Figure 13 represents the update of the hot data metadata according to the communication cycle. The client #1 organically communicates with the load balancer at every communication cycle ($\tau$) and updates its hot data metadata information. The communication cycle ($\tau$) with the load balancer for the hot data update of the client is determined according to the update rate of the metadata updated by the load balancer. If the update rate of the client exceeds the threshold value, the load balancer updates the metadata of the client. This is because the old metadata information held by the client is replaced with the up-to-date metadata information of the load balancer. If the update rate is lower than the threshold value, the metadata information of the load balancer is updated. As the clients and the load balancer update and synchronize hot data metadata through an organic communication cycle, when client #1 requests $D_1$, it can directly access the node and obtain $D_1$ without going through the load balancer, thereby reducing the unnecessary processing time.

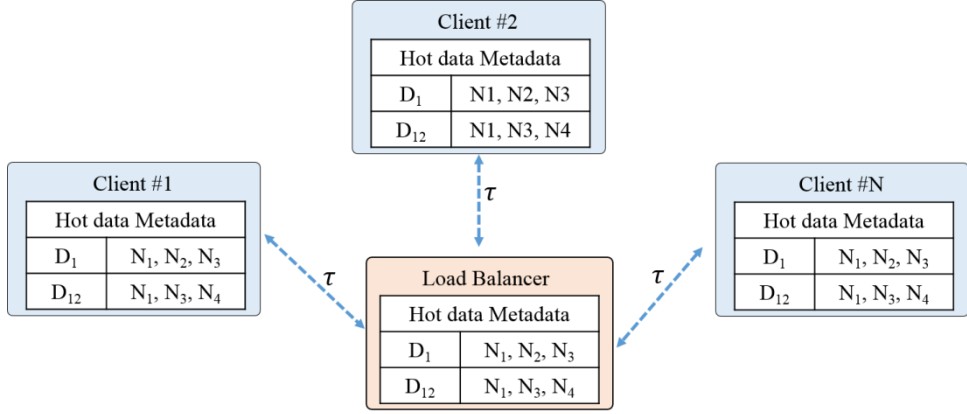

**Figure 13.** Update of the hot data metadata according to the communication cycle.

If the organic cache metadata update policy is not applied, when the user requests $D_1$, client #1 retrieves its metadata and accesses the node to obtain $D_1$. If $D_1$ is not present in the node due to data replacement, however, client #1 must communicate with the load balancer to determine the location where $D_1$ is stored and request $D_1$. When this problem occurs, it takes a long time to access the data if the hot data metadata of the load balancer and that of the client are not synchronized.

Figure 14 shows the algorithm for the hot data metadata synchronization of the client. The load balancer updates the metadata for the hot data and calculates the update rate of the metadata through Equation (10) at each communication cycle. At this time, t represents the time of updating the metadata, $UR_t$ represents the renewal rate at time t of the metadata, $NUM_t$ represents the number of metadata updated at time t, and $NM_t$ represents the number of metadata held by the client at time t. If the renewal rate is above the threshold, the client's metadata through metadata synchronization is updated

---

**Client_hot_data_metadata_update**

---

01 :   *while(communication cycle)*

02 :    *if(the hot data is changed)*

03 :     *update metadata of hot data by load balancer;*

04 :    *calculate $UR_t$ of metadata at time t;*

05 :    *if ($UR_t$ > threshold value)*

06:     *update the metadata of the clients through metadata synchronization ;*

---

**Figure 14.** Client hot data metadata synchronization algorithm.

$$UR_t = \frac{NUM_t}{NM_t} \tag{10}$$

## 4. Performance Evaluation

### 4.1. Evaluation Environment

To verify the effectiveness of the proposed load balancing scheme, various performance evaluations were conducted using the existing load balancing schemes. For the performance evaluation, a simulation was performed using a Java program on a PC using an Intel(R) Core(TM) i5-4440 CPU, 3.10 GHz, RAM 8.00 GB, and Microsoft Window 7 64-bit operating system. Table 1 shows the performance evaluation parameters. The number of initial nodes was 4–16, the memory size of each node was 16–256 MB, the number of replicas was 3, the load of the nodes was 30–80, and the number of data requests was 1000–5000. For convenience, the scheme proposed in [33] is called Adaptive Performance Aware (APA). The APA calculates the costs of the nodes based on their hit rates and usage rates, and distributes the load by adjusting the hash spaces based on the calculated values [33]. The R-Memcached considers the hot data that causes a high load on a node and distributes the load by replicating the hot data to other nodes [34]. Both schemes distribute the load considering only one feature when the load is unbalanced. The performance evaluation was conducted using APA and R-Memcached, which are considered as representative load balancing schemes.

**Table 1.** Performance evaluation parameters.

| Parameter | Value |
| --- | --- |
| # of initial nodes (n) | 4–16 |
| memory size (MB) | 16–256 |
| # of replicas (n) | 3 |
| the load of the nodes | 30–80 |
| # of queries (n) | 1000–5000 |

To verify the success of the proposed scheme, the performances, including the load balancer access frequency, the node access frequency, and the query execution time, were compared with APA and R-Memcached. The number of query requests and the number of nodes were changed during this comparison. In the performance evaluation, the query processing time was measured while the number of nodes and the number of query requests were changed. The load balancer access and node access frequencies were measured while the number of query requests was changed.

### 4.2. Self-Performance Evaluation

The existing load balancing schemes distribute the load using either the replication method or the migration method. When only one method was used, it could not solve the overload problems caused by various issues. Therefore, in order to prove that the scheme that uses both replication and migration is superior to the schemes that use only one method, a performance comparison was carried out. For the performance comparison using the proposed scheme, the query processing time was measured using replication, migration, and replication+migration while the number of nodes was changed. Figure 15 shows the measured query execution times according to the number of nodes. When the number of queries is 5000, the execution time was measured while the number of nodes was changed. Since the load balancing is performed using either replication or migration, load imbalance problems that occur in various situations could not be addressed. The load balancing scheme, which combines data replication and migration, addresses the load imbalance due to node performance and hot data, thus providing better performance than schemes using only data replication or data migration. In terms of the query processing time, the replication+migration scheme was improved by up to 21% over data migration only and by up to 16% over data replication only.

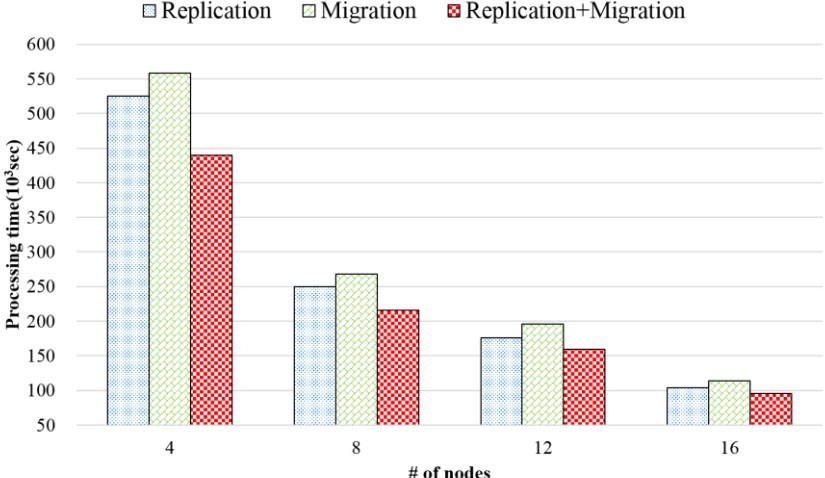

**Figure 15.** Query processing time according to the number of nodes.

To demonstrate the effectiveness of the scheme that uses replication+migration even when the number of query requests is changed, the query processing time was measured using replication, migration, and replication+migration. Figure 16 shows the measured query execution times according

to the number of query requests. When the number of initial nodes is 8, the execution time was measured while the number of query requests was changed. As the proposed scheme combines replication and migration to address the load imbalance problems that occur due to various causes, such as hot data or node performance, it showed better performance than the schemes that use either replication or migration even if the number of query requests was changed. The results of the performance evaluation revealed that the scheme that used replication+migration showed a maximum 15% reduction in the query processing time compared to the scheme that used migration, and a maximum 13% reduction compared to the scheme that used replication.

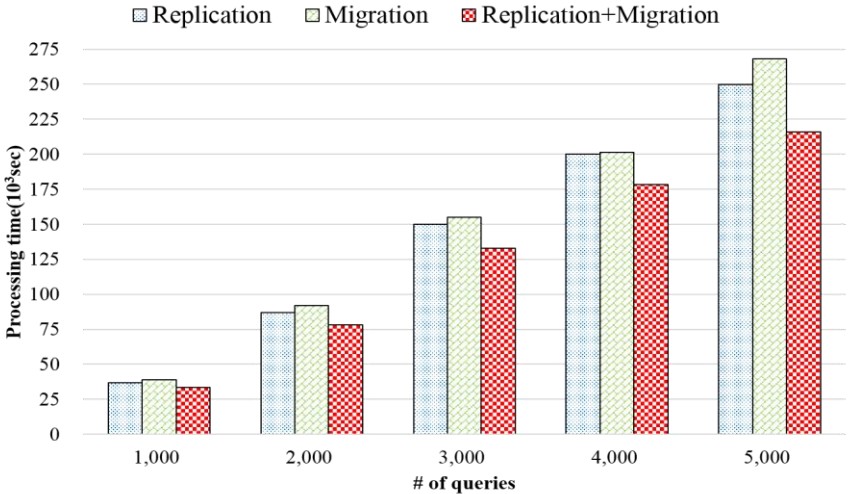

**Figure 16.** Query processing time according to the number of query requests.

*4.3. Load Balancer Access Frequency*

The existing schemes obtained data from the nodes through the load balancer. When the load is concentrated on the load balancer, however, the load balancer is overloaded. Therefore, it is important to reduce the load on the load balancer. The load balancer access frequency was measured to show that the access to the load balancer was reduced. To compare the load balancer access frequencies according to the user requests, the load balancer access frequency was measured according to the user requests. Figure 17 shows the measured load balancer access frequencies according to the number of query requests. When the number of initial nodes is 8, the load balancer access frequency was measured while the number of query requests was changed between 1000 and 5000. For all cases in which APA and R-Memcached users request data, the corresponding node is accessed through the load balancer. If the load is concentrated on the load balancer, the load balancer can be overloaded, lowering the overall performance. In the proposed scheme, however, the load balancer manages the hot data metadata, and periodically synchronizes the metadata with the clients. Therefore, when a client accesses the hot data, it directly accesses the node without going through the load balancer, and rapidly requests the data. It also reduces access to the load balancer to prevent the overload of the load balancer. As the amount of hot data increases with the number of query requests, the load balancer access frequency decreases further. The results of the performance evaluation revealed that the proposed scheme showed a maximum 24% reduction in the load balancer access frequency compared to APA and R-Memcached.

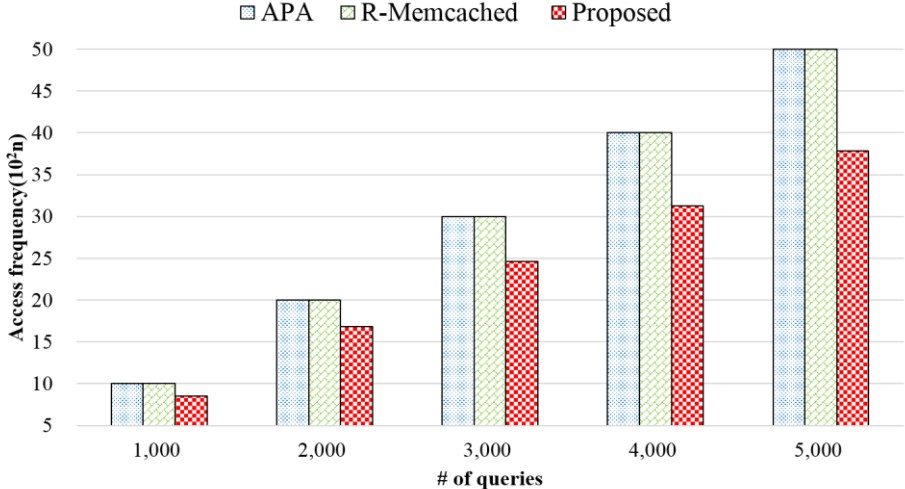

**Figure 17.** Load balancer access frequency according to the number of query requests.

To demonstrate that the load balancer access frequency decreases even when the number of nodes is changed, a performance evaluation was conducted using the existing schemes while the number of nodes was changed. Figure 18 shows the measured load balancer access frequencies according to the number of nodes. As the number of nodes increased, the overall memory size increased, and a large amount of data could be stored. In APA and R-Memcached, the load balancer manages all metadata, so the load balancer needs to be accessed each time for data access. As a result, an increase in the number of nodes does not change the rate of access to the load balancer. The proposed scheme compares the update rate of the metadata and if the metadata changes above the threshold, the client updates the metadata through synchronization with the load balancer. As the number of nodes increases, the stored data increases, which increases the change in hot data. Therefore, as the number of nodes increases, the client stores the latest metadata, which can reduce the access frequency of the load balancer. The results of the performance evaluation showed that the proposed scheme demonstrates a maximum 33% reduction in the load balancer access frequency compared to APA and R-Memcached.

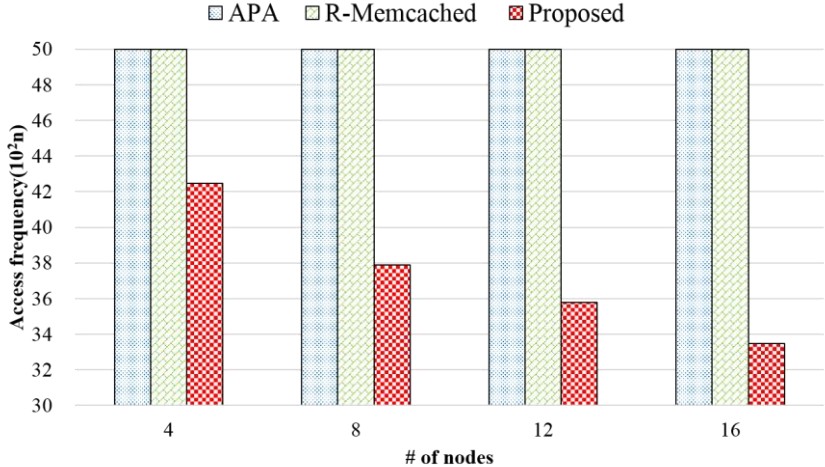

**Figure 18.** Load balancer access frequency according to the number of nodes.

### 4.4. Node Access Frequency

If the load is concentrated on a specific node, the node is overloaded and the performance of the system is degraded. Therefore, it is important to evenly distribute the load to the nodes. To show that the access is evenly distributed to the nodes, the access frequency was measured while the number of query requests were changed. To compare the node access frequencies according to the number

of query requests, the node access frequencies were measured and their standard deviations were calculated. The node access frequency means the number of memory accesses when processing a query in the node. Figure 19 shows the measured node access frequencies according to the number of query requests. The node access frequency was measured while the number of query requests was changed between 1000 and 5000. Figure 19a shows the node access frequency when performing 1000 queries. The difference in the number of accesses by the node was 87 for APA and 99 for R-Memcached, which gave APA good load balancing performance. The proposed scheme achieved the best load balancing performance over the existing schemes since the maximum difference of the access frequency by the node is 54. Figure 19b–e show the node access frequency when performing 2000 queries, 3000 queries, 4000 queries, and 5000 queries, respectively. The performance evaluation results of Figure 19b–e are similar to those of Figure 19a. As a result, the performance evaluations according to the number of query requests resulted in APA achieving a better performance than R-Memcached and the best load balancing performance of the proposed scheme. In the case of APA, the costs of the nodes are calculated and the hash spaces of the nodes are adjusted using the calculated values. When the load of the neighbor node is high, however, the overload problem cannot be addressed because the adjustment amounts of the hash spaces are low. Furthermore, when hot data exists, a large amount of data is migrated, causing a considerable migration cost. R-Memcached replicates the hot data to two of the successor nodes to distribute the load. When a node is deleted, however, overlapping hot data can be stored in one node, thereby rapidly increasing the load on a specific node again. In addition, this scheme does not propose any load balancing method for cases where hot data does not exist in the overloaded node. Therefore, when there is no hot data, the load cannot be distributed. In the proposed scheme, however, when an overloaded node occurs, load balancing methods are proposed according to the situations. If there is any hot data, it divides the hash space evenly and replicates the hot data evenly to the nodes, thereby preventing overlapping hot data from being stored in one node when a node is deleted. If there is no hot data, the load is distributed by adjusting the hash spaces based on the loads of the nodes.

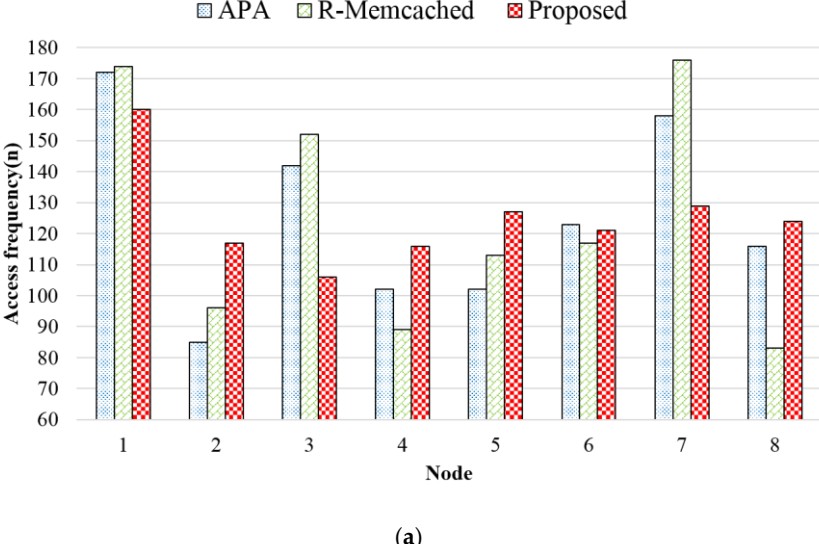

(**a**)

**Figure 19.** *Cont.*

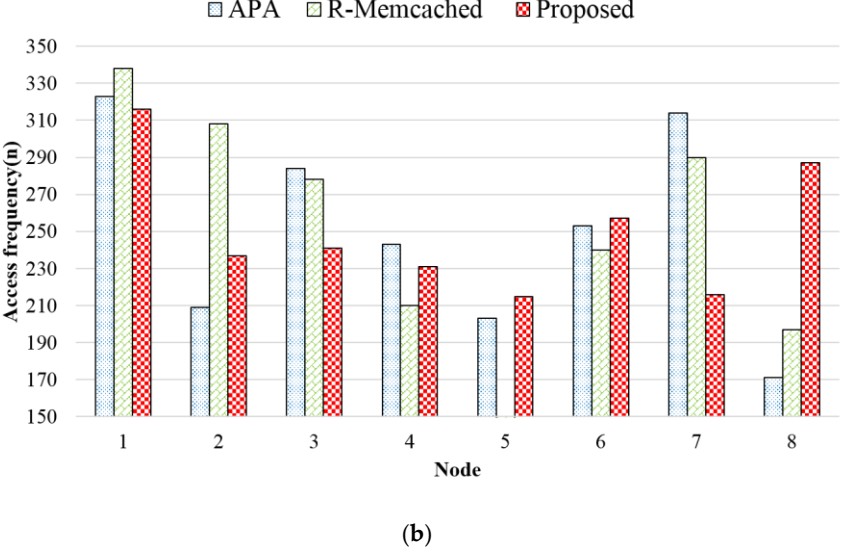

(**b**)

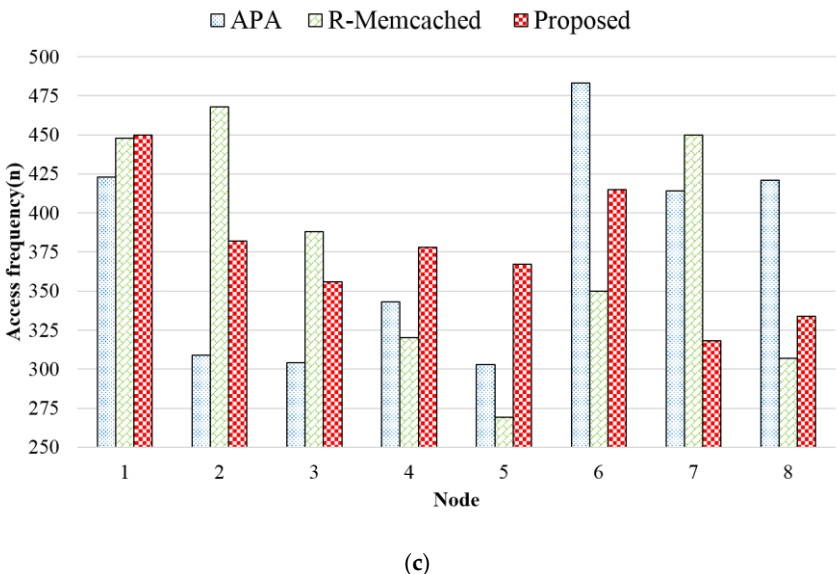

(**c**)

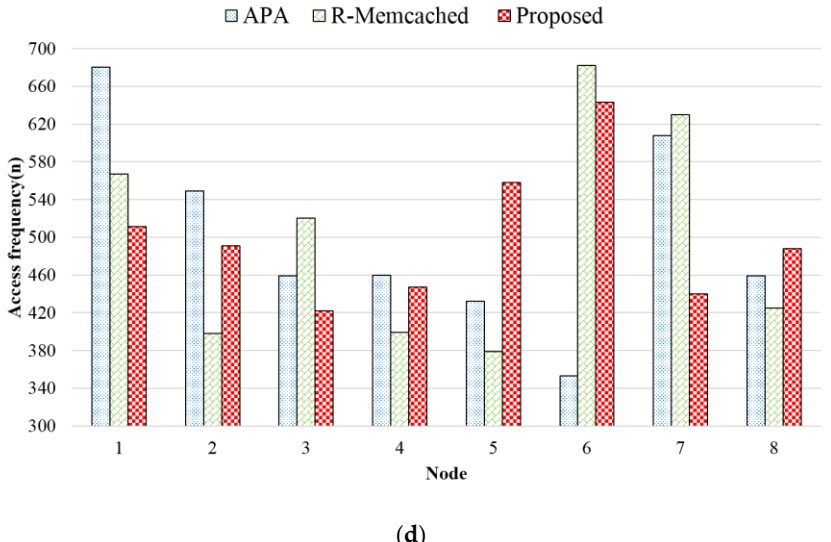

(**d**)

**Figure 19.** *Cont.*

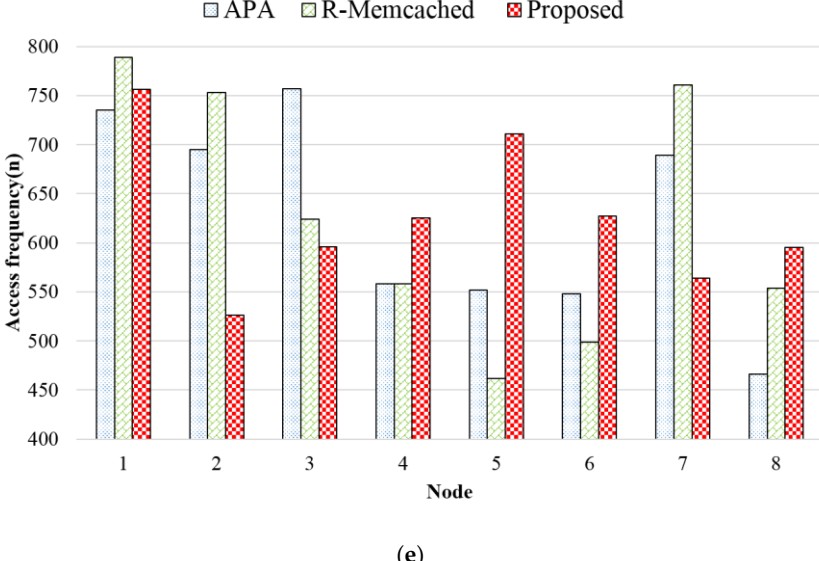

(**e**)

**Figure 19.** Node access frequency according to the number of query requests. (**a**) 1000 queries. (**b**) 2000 queries. (**c**) 3000 queries. (**d**) 4000 queries. (**e**) 5000 queries.

Figure 20 compares the standard deviations of the node access frequency according to the number of query requests. As shown in the figure, the standard deviations of the node access frequency using the proposed scheme were the lowest compared to the existing schemes. Therefore, it can be seen that the proposed scheme has more uniform node access than the existing schemes. The results of the performance evaluation revealed that the proposed scheme showed a maximum 47% reduction in the standard deviation of the node access frequency compared to APA and a maximum 57% reduction compared to R-Memcached.

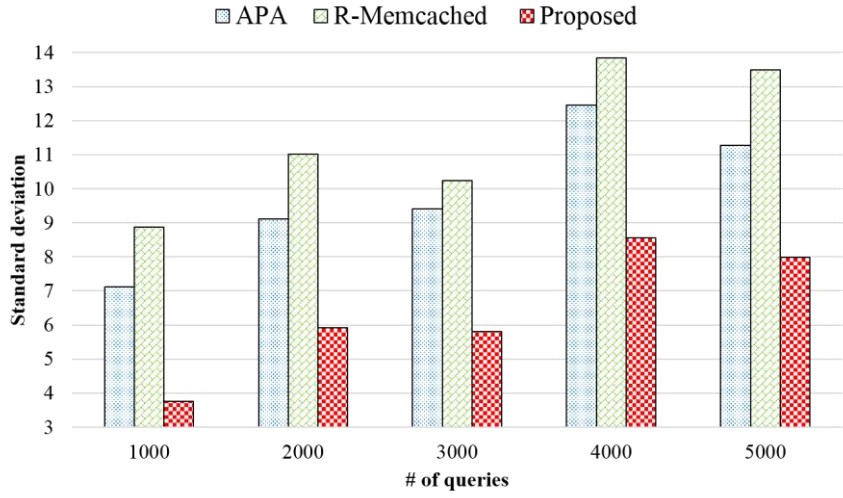

**Figure 20.** Standard deviation of the node access frequency according to the number of query requests.

### 4.5. Query Processing Time

It is important to process user queries rapidly in distributed systems. To demonstrate that the proposed scheme has a better query processing time than the existing schemes, a performance evaluation was conducted while the number of nodes and the number of query requests were varied. To compare the query processing time between the existing schemes and the proposed scheme, the query processing time was measured while the number of query requests was changed. Figure 21 shows the query processing time according to the number of query requests. The evaluation was performed while the number of query requests was varied from 1000 to 5000. As mentioned in the previous performance evaluation, unlike APA

and R-Memcached, the clients retain the metadata information about the hot data in the proposed scheme and they request the hot data directly to the node without going through the load balancer, resulting in the fast node access speed. When an overloaded node occurs, APA uses the data migration method and R-Memcached uses the data replication method. The proposed scheme, however, analyzes the root-cause of the overloaded node. Based on the cause, if any hot data exists in the overloaded node, the data replication method is used. If hot data does not exist, the data migration approach is used. The results of the performance evaluation revealed that the proposed scheme showed a maximum 23% reduction in the query processing time compared to APA, and a maximum 18% reduction compared to R-Memcached.

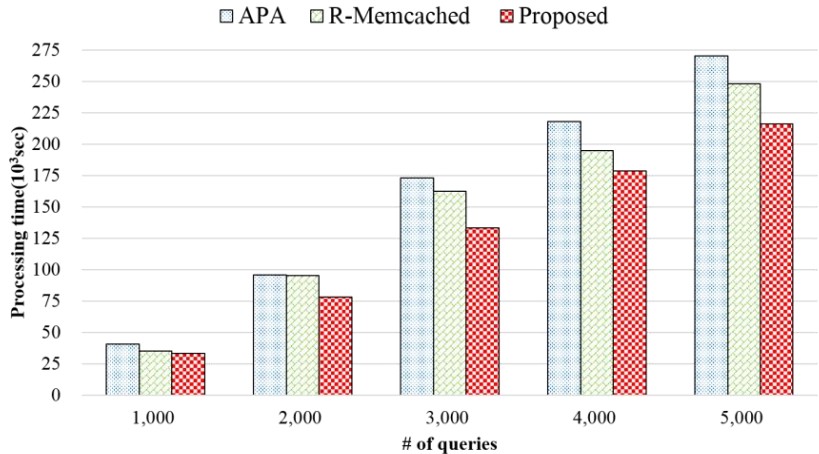

**Figure 21.** Query processing time according to the number of query requests.

To demonstrate the performance of the proposed scheme even when the number of nodes is changed, the query processing time was measured according to the number of nodes. Figure 22 shows the query processing time according to the number of nodes. The evaluation was performed using 5000 query requests. As mentioned earlier, unlike APA and R-Memcached, the clients in the proposed scheme request the hot data directly to the node without going through the load balancer, thereby rapidly accessing the data node. In addition, based on the cause of the overload, if any hot data exists in the overloaded node, the data replication method is used. If hot data does not exist, the data migration method is used. The proposed scheme showed a reduced query processing time compared to the existing schemes even when the number of nodes was changed. The results of the performance evaluation revealed that the proposed scheme showed a maximum 20% reduction in the query processing time compared to APA, and a maximum 14% reduction compared to R-Memcached.

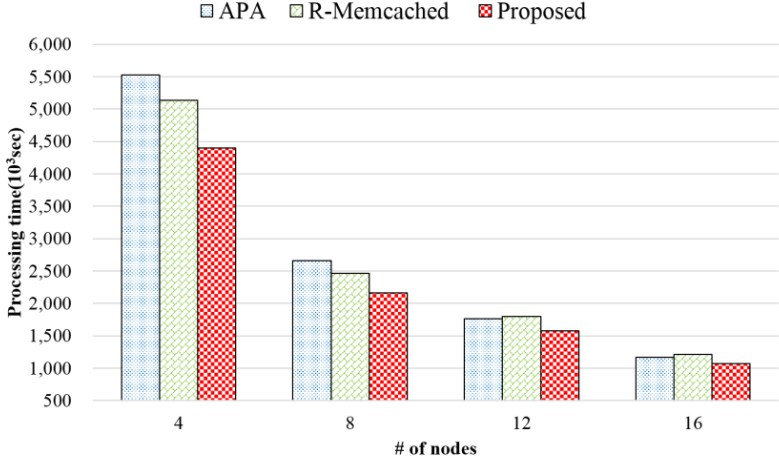

**Figure 22.** Query processing time according to the number of nodes.

## 5. Conclusions

In this paper, we proposed a load balancing scheme considering the load characteristics of the nodes in distributed in-memory environments. The proposed scheme allocates the initial nodes considering the memory of the nodes. If an overloaded node occurs and hot data exists, it evenly divides the hash space and replicates the hot data to each divided hash space. If hot data does not exist, the hash space is adjusted based on the loads of the nodes. In addition, when a node is added or removed in a cluster environment, the load is distributed by adjusting the hash spaces considering the loads of the predecessor node and the successor node. Finally, the clients retain the metadata of the hot data and periodically update the metadata by communicating with the load balancer, thereby increasing the access efficiency to the hot data. This improves the load balancing performance of each node, and thus, the load of each node can be distributed more effectively, and the node overload can be reduced. As a result, the problem of the load being concentrated on a specific node due to popular data or data that attracts people's attention can be addressed more efficiently. This scheme can be used to perform load balancing in structured file sharing systems that use memory or streaming services. The results of the performance evaluation confirmed that the proposed scheme showed better performance in the query processing time than the existing schemes. Compared to the existing schemes, the load balancer access frequency was reduced by a maximum of 24%, and the standard deviation of the node access frequency was decreased by a maximum of 57%. The query processing time was reduced by a maximum of 23%.

**Author Contributions:** Conceptualization, K.B., K.C., D.C., J.L. and J.Y.; methodology, K.B., K.C., D.C. and J.L.; validation, K.C.; writing—original draft preparation, K.B. and K.C.; supervision, J.Y.; writing—review and editing, J.Y.

**Funding:** This work was supported by Institute of Information & Communications Technology Planning & Evaluation (IITP) grant funded by the Korea government (MSIT) (No.B0101-15-0266, Development of High Performance Visual BigData Discovery Platform for Large-Scale Realtime Data Analysis), by Next-Generation Information Computing Development Program through the National Research Foundation of Korea (NRF) funded by the Ministry of Science, ICT (No. NRF-2017M3C4A7069432), and by "Human Resources Program in Energy Technology" of the Korea Institute of Energy Technology Evaluation and Planning (KETEP), granted financial resource from the Ministry of Trade, Industry & Energy, Republic of Korea. (No. 20164030201330).

**Conflicts of Interest:** The authors declare no conflict of interest.

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
