# Peer review of "Load Balancing Scheme for Effectively Supporting Distributed In-Memory Based Computing"

_electronics, doi:10.3390/electronics8050546_

Round 1

Reviewer 1 Report

This is an interesting work, which presents a load balancing scheme for distributed in-memory. The presentation is nice, the results presented are quite convincing and the literature review is complete. My opinion is that it is worth of publication, as it is. 

Author Response

Please refer to the attached PDF file.

Reviewer 2 Report

The paper presents an interesting scheme for load balancing with a good description and evaluation. However, the main contribution should be specified clearly to be distinguished from the authors previous work presented in the conference paper as mentioned [37].

All comments and suggestions can be found in the attached file.

Author Response

Please refer to the attached PDF file

Round 2

Reviewer 2 Report

All suggestions and comments almost have been considered